# Microbiological Quality of Nuts, Dried and Candied Fruits, Including the Prevalence of *Cronobacter* spp.

**DOI:** 10.3390/pathogens10070900

**Published:** 2021-07-15

**Authors:** Anna Berthold-Pluta, Monika Garbowska, Ilona Stefańska, Lidia Stasiak-Różańska, Tamara Aleksandrzak-Piekarczyk, Antoni Pluta

**Affiliations:** 1Department of Technology and Food Assessment, Division of Milk Technology, Institute of Food Sciences, Warsaw University of Life Sciences—SGGW, 02-787 Warsaw, Poland; monika_garbowska@sggw.edu.pl (M.G.); lidia_stasiak@sggw.edu.pl (L.S.-R.); antoni_pluta@sggw.edu.pl (A.P.); 2Department of Preclinical Sciences, Institute of Veterinary Medicine, Warsaw University of Life Sciences—SGGW, 02-787 Warsaw, Poland; ilona_stefanska@sggw.edu.pl; 3Institute of Biochemistry and Biophysics, Polish Academy of Sciences (IBB PAS), 02-106 Warsaw, Poland; tamara@ibb.waw.pl

**Keywords:** *Cronobacter* spp., nuts, dried fruits, seeds, ready-to-eat foods, food control

## Abstract

*Cronobacter* genus bacteria are food-borne pathogens. Foods contaminated with *Cronobacter* spp. may pose a risk to infants or immunocompromised adults. The aim of this study was to determine the microbiological quality of nuts, seeds and dried fruits with special emphasis on the occurrence of *Cronobacter* spp. Analyses were carried out on 64 samples of commercial nuts (20 samples), dried fruits (24), candied fruits (8), seeds (4), and mixes of seeds, dried fruits and nuts (8). The samples were tested for the total plate count of bacteria (TPC), counts of yeasts and molds, and the occurrence of *Cronobacter* spp. *Cronobacter* isolates were identified and differentiated by PCR-RFLP (Polymerase Chain Reaction - Restriction Fragments Length Polymorphism) and RAPD-PCR (Random Amplified Polymorphic DNA by PCR) analysis. TPC, and yeasts and molds were not detected in 0.1 g of 23.4%, 89.1%, and 32.8% of the analyzed samples. In the remaining samples, TPC were in the range of 1.2–5.3 log CFU g^−1^. The presence/absence of *Cronobacter* species was detected in 12 (18.8%) samples of: nuts (10 samples), and mixes (2 samples). The 12 strains of *Cronobacter* spp. included: *C. sakazakii* (3 strains), *C. malonaticus* (5), and *C. turicensis* (4). The results of this study contribute to the determination of the presence and species identification of *Cronobacter* spp. in products of plant origin intended for direct consumption.

## 1. Introduction

Nuts and dried fruits belong to the category of low moisture food (LMF) [1]. According to the FAO/WHO definition [1], this food category includes food products with a naturally low moisture content and products in which the moisture content has been reduced through drying or dehydration. Low moisture food products are characterized by extended stability owing to their low water activity (a_w_). Ntuli et al. [2] and Danyluk et al. [3] reported that a_w_ of commercially dried fruits and nuts was approximately 0.78–0.83 and less than 0.70, respectively. Such a level of a_w_ prevents growth of many pathogens and lends to the microbiological safety of the LMFs. Although bacteria of the Enterobacteriaceae family (*Escherichia coli, Salmonella*) cannot propagate on plant products with a low a_w_, they can remain alive on/in such products for a long time (up to 1 year), especially when products are stored in cold [3,4].

Nuts are used by the food industry as they are considered a healthy, convenient, and nutritious snack. Therefore, they are an important element of the development of the bakery and confectionery industry. Data published by the Food and Agriculture Organization of the United Nations (FAOSTAT) shows that the largest producers of walnuts in 2019 were China, USA and Iran (total world production in 2019 amounted to 4.5 million tons). The global production of hazelnuts and almonds in 2019 was 1.1 million tons and 3.5 million tons, respectively. In turn, the production of Brazilian nuts in the world was significantly lower (70.2 thousand tons) and their largest producers are South America countries (Brazil, Bolivia and Peru) [5].

It is known that fruits are carriers of non-pathogenic and pathogenic microflora, and they can become contaminated at the initial stages of their acquisition (growth, harvesting), during their processing and distribution, as well as improper handling [2,6]. Contrary to the widely reported occurrence of pathogens on fresh fruits and food poisoning caused by the consumption of contaminated fresh fruits [7,8], the presence of pathogens on dried or candied fruits is not well documented [2]. Uncommonly linked to foodborne illness, several nuts have been associated with outbreaks of salmonellosis or *E. coli* O157:H7 gastroenteritis [9,10,11].

*Cronobacter* spp. bacteria are associated with rare but life-threatening diseases of newborns and infants (necrotizing enterocolitis, meningitis and sepsis) [12,13], but also of elderly and immunocompromised adults (septicemia, pneumonia, osteomyelitis, splenic abscesses, and wound infections) [14,15]. The *Cronobacter* genus consists of seven species: *C. sakazakii*, *C. turicensis*, *C. malonaticus*, which are the most frequently isolated from human infections, *C. condimenti*, *C. dublinensis*, *C. muytjensii* and *C. universalis* [16,17,18,19].

Many researchers have focused on the control of *Cronobacter* spp. in powdered infant formulas [20,21,22]. These microorganisms have also been isolated from different food products: cereals [23,24,25], fruits and vegetables [26,27,28,29], herbs and spices [30,31], milk products [32,33], meat [34,35], fish and products made from them [36] and ready-to-eat products [37].

Literature data indicate that the source of food contamination by *Cronobacter* spp. may be the raw material (especially in plant products) and the environment of food production [38]. *Cronobacter* genus bacteria demonstrate a whole range of properties which enable them to survive in various food products and allow for their high ability of adaptation to a changing environment during the manufacturing process. Among the key characteristics of *Cronobacter* spp. is, first of all, their resistance to desiccation (low a_w_). High resistance to desiccation results most probably from trehalose concentration in *Cronobacter* cells (in particular those in stationary phase) which acts as a protective factor [39]. It is believed that the mechanism of a hyper-osmotic stress response system may involve the substitution of the water layer that accumulates around the biomolecules by trehalose, which enables maintenance of the three-dimensional structure of these molecules (“the water replacement theory”) [40]. Another trait allowing the bacteria of *Cronobacter* genus to colonize in the manufacturing environment, which is also an important attribute related to their virulence, is their ability to form biofilms which provide protection against environmental stress factors and also against the factors of the host’s immune response [41]. Numerous studies have demonstrated *C. sakazakii* to be capable of adhering to the surface of materials used in the food manufacturing industry in contact with food, such as silicon, latex, polycarbonate, stainless steel, glass, polyvinyl chloride, which may be affected by the production of extracellular polysaccharides (EPS) by this species [42,43,44,45]. The ability of *C. sakazakii* to form biofilms on a variety of surfaces makes it easier for them to contaminate food in the production environment. This applies to the entire food-processing chain. Biofilms are difficult to remove, contaminating food and posing a potential health risk to consumers.

Iversen and Forsythe [46] were the first ones to formulate the hypothesis that the natural ecosystem of the *Cronobacter* genus bacteria were plants. It was based on several physiological traits of these microorganisms: production of a extracellular polysaccharides, production of yellow pigment (carotenoids) and their extreme desiccation resistance. These traits may enable the bacteria to colonize the plant environment, offer protection against oxygen radicals generated by sunlight and help to survive dry periods.

There are no regulations for *Cronobacter* spp. in foodstuffs except for the powdered infant formulas. There is also no published evidence for foodborne disease causation by this group of bacteria from other types of food. However, some data has recently appeared which suggests that recently there may be other food sources of *Cronobacter* infections in the elderly persons, hence the plant-origin food products cannot be excluded [12,47]. According to some speculations, the difficult identification of these pathogens due to the lack of appropriate diagnostic methods in many laboratories, causes an underestimation of the number of *Cronobacter* spp. infection cases (e.g., in the elderly) [48].

This work was aimed at studying the microbiological quality of commercial ready-to-eat food products of plant origin (nuts, seeds, dried and candied fruits), and at determining the prevalence of *Cronobacter* spp. in these products with the use of PCR-RFLP (Polymerase Chain Reaction - Restriction Fragments Length Polymorphism), and RAPD-PCR analysis (Random Amplified Polymorphic DNA by PCR) for identification/differentiation of isolates.

## 2. Results and Discussion

### 2.1. Prevalence of Total Plate Count (TPC) of Bacteria, Yeasts, and Molds in Tested Samples

Results of total plate count (TPC) of bacteria in the tested products are summarized in Table 1.

Regarding the TPC, 23.4% of the samples had an total plate count below 1 log CFU (Colony Forming Units) g^−1^ (TPC were not detected in 0.1 g of 20.0%, 41.7%, and 12.5% of the samples of nuts, dried fruits, mixes, respectively, and in any of the samples of seeds and candied fruits). In about 18.8% of the samples, the count of these bacteria ranged from >1 to 2 and in the same percentage of samples it was from >3 to 4 CFU g^−1^, but in most of the samples (31.3%) the TPC count was in the range from >2 to 3 log CFU g^−1^. Only in 2 samples of dried fruits did the total plate count of bacteria exceed 5 log CFU g^−1^. In the samples of nuts and seeds, the TPC count did not exceed 5 log CFU g^−1^ in any of the samples, whereas in the samples of candied fruits it did not exceed 3 log CFU g^−1^. The lowest mean of TPC was found in the samples of candied fruits (2.20 log CFU g^−1^), while the highest in the samples of dried fruits and seeds (3.07 and 3.17 log CFU g^−1^, respectively) (Table 1). These results were comparable to those reported by Danyluk et al. [3], Eglezos et al. [49], Davidson et al. [9], and Harris et al. [50] for nuts, but were slightly higher than these provided by Ntuli et al. [2] for dried fruits.

The level of TPC in nuts is believed to be indicator of postharvest contamination of these products (e.g., with soil) and, indirectly, the presence of harmful lipolytic bacteria [49]. Davidson et al. [9] reported TPC ranging from 1.0 to 5.4 log CFU g^−1^ in walnuts, however the greatest percentage of the samples (37%) were those with total plate bacteria count at 3–4 log CFU g^−1^. In 45–87% of the samples of different kinds of pre-roasted nuts received into Australian nut-processing facilities, the mean aerobic plate counts were from 2.5 to 4.5 log CFU g^−1^ [49]. In none of the commercial samples of various kinds of nuts was the aerobic bacteria count higher than 4 log CFU g^−1^ [51]. In turn, TPC in raw pistachios ranged from 2 to 5.4 log CFU g^−1^ [50].

Results of determinations of the yeast and molds counts in the analyzed samples of nuts, dried and candied fruits, and mixes thereof, are summarized in Table 2 and Table 3.

In our study, yeasts and molds were not detected in 0.1 g of 89.1% and 32.8% of the samples, respectively. Yeasts were not detected in any sample of seeds and candied fruits (in 0.1 g). There were no statistically significant differences in the count of yeasts in the samples of different product groups (Table 2). In turn, contamination of the samples with molds was significantly greater than with yeast and reached >3–4 log CFU g^−1^ in 2 samples of dried fruits (Goji berries and prunes), 1 sample of seeds (pumpkin seeds), and 2 samples of mixes. The highest mean count of molds was found in the samples of “mixes” (3.22 log CFU g^−1^), and the lowest in nuts (1.99 log CFU g^−1^) and candied fruit (1.5 log CFU g^−1^) (Table 3). While in our study fungi were determined in less than 67% of the dried fruit samples, Witthuhn et al. [52] reported their occurrence in almost all samples of dried fruits examined in Africa. Yeasts and molds were detected in 6.7% and 100% of the walnut samples, with counts ranging from 1.0 to 5.2 and from 1.0 to 5.0 log CFU g^−1^, respectively [9]. In raw almonds, yeasts were not detected (<10 CFU g^−1^) in 89% of the samples and mold levels ranged from 1.0 to >5.2 log CFU g^−1^, with average counts of 3.5 log CFU g^−1^ [3]. Likewise, yeast counts were below the limit of detection (<1 log CFU g^−1^) and mold counts ranged from < 1 to 4.1 log CFU g^−1^ in 96% of the samples of raw pistachios analyzed by Harris et al. [50].

Following recommendations for dried fruits, the total counts of yeasts and molds should not exceed 3 log CFU g^−1^ [2,52]. All samples analyzed in our study met those guidelines regarding contamination with yeasts, whereas five samples (7.8%) failed to meet them in terms of contamination with molds.

### 2.2. Presence of Cronobacter spp. in the Tested Food Products and Phenotypic Identification of Presumptive Isolates

Twelve samples out of 64 tested (18.8%) classified as nuts and mixtures of seeds, dried fruit and nuts, on the basis of phenotypic characteristics, were considered contaminated with bacteria of the genus *Cronobacter*, and 12 isolates were isolated from them (one isolate from each sample: almonds, hazelnuts, Brazil nuts, cashews, pine nuts, macadamia nuts and a mixture of dried fruit, seeds and nuts). *Cronobacter* spp. were not detected in any of the analyzed samples of dried fruits, candied fruits, and seeds. All the 12 isolates showed the typical biochemical profile according to ISO-22964:2017 [53] and gave yellow-pigmented colonies on tryptone soya agar (TSA) plates at 25 °C for 48 h.

### 2.3. Identification of the Isolates from the Genus Cronobacter

Two restriction digestions of the 659-bp fragment of the *rpo*B gene were performed to enable species identification of isolates (Figure 1). The results of PCR-RFLP using a single (*Hin*P1I) and double restriction enzyme simultaneously (*Hin*P1I and *Csp*6I) showed that three isolates belonged to *C. sakazakii* species. The 9n and 10n isolates had identical RFLP patterns, corresponding to *C. sakazakii* NCTC 8155, and the 11m isolate showed different fragments, identical to that of *C. sakazakii* ATCC 29544 type strain (Table 4). The RFLP analysis of the nine remaining isolates with *Hin*P1I resulted in restriction profiles common to *C. turicensis* and *C. malonaticus* species (fragments of 312, 186 and 142 bp in size). Further double digestion of these isolates with *Hin*P1I and *Csp*6I simultaneously, resulted in clearly different patterns as *C. turicensis* had a 206 bp fragment, whereas *C*. *malonaticus* had a fragment of 167 bp in size. On this basis, four strains (1n, 2n, 3n and 4n) were identified as *C. turicensis* and five as *C. malonaticus* (5n, 6n, 7n, 8n and 12m) (Table 4).

RAPD-PCR with three different 10-nt random primers was used for intra-species differentiation of all isolates (Figure 2). RAPD profiles with primer UBC245 and UBC282 revealed a total of five patterns, whereas the RP primer enabled additionally differentiating one of the isolates (Table 5). Three *C. turicensis* strains had identical RAPD profiles, which suggests that they were clonally related. Strain 2n presented a clearly distinct profile only with the RP primer. Among the *C. malonaticus* species, the isolates 5n, 6n, 7n and 8n were indistinguishable (isolates from samples of products originating from the same manufacturer), while profiles of the 12m strain contained bands of various sizes. In the case of *C. sakazakii* isolates, the results of RAPD and PCR-RFLP were consistent and indicated that the 9n and 10n isolates revealed identical profiles and should be considered as the same strain. In turn, *C. sakazakii* 11m showed distinct patterns, regardless of the method used.

In our previous work, 13 RAPD patterns in 21 *Cronobacter* spp. isolates were identified, whereas the 12 strains isolated in this study were less differentiated and revealed 6 different patterns [26]. In both studies RAPD-PCR methods with RP, UBC245 and UBC282 primers were shown as a valuable and reliable tool in genetic differentiation of *Cronobacter* spp. strains. Other molecular methods used to differentiate *Cronobacter* are DNA-sequence-based genotyping, including multilocus sequence typing (MLST) of 7 housekeeping genes (7-loci MLST), ribosomal-MLST (r-MLST, 53 loci), core genome-MLST (cg_MLST, 1836 loci) and CRISPR (clustered regularly interspersed short palindromic repeat)-*cas* array profiling [15,54]. Most of these molecular typing methods based on in silico analysis of whole genome sequences and provide interspecific and intraspecific differentiation between highly clonal *Cronobacter* species, including phylogenetic and evolutionary analysis studies, epidemiological and environmental investigation of strains as well as a better understanding of their virulence potential. In recent years, the availability of NGS technology has expanded for academic and industry institutions and microbiological laboratories, and the costs of NGS testing have dropped significantly. Nevertheless, the use of whole-genome sequencing-based typing is still expensive and technically difficult and, therefore, rapid and simple PCR-based methods, e.g., PCR-RFLP or RAPD-PCR, are required and still widely used for identification and differentiation of isolates from foods [24,26,55,56]. It is interesting to note that the differentiation of *Cronobacter* spp. strains based on the PCR-RFLP and RAPD-PCR methods used in this study in the case of the isolates from sprouts, vegetables, and non-pasteurized juices gave the results completely consistent with the results of the MLST analysis (unpublished data).

To recapitulate, the presence of *C. sakazakii* bacteria was confirmed in 10.0% of the nut samples (Brazilian nuts) and in 12.5% of the samples of mixes of dried fruits, seeds and nuts; in turn *C. turicensis* was detected in 20.0% of the nut samples (almonds and hazelnuts); while *C. malonaticus* in 20.0% of the nut samples (hazelnuts, cashew, pine nuts, macadamia nuts) and in 12.5% of the samples of mixes of dried fruits, seeds and nuts (Table 6). Literature data concerning the prevalence of *Cronobacter* spp. in nuts, seeds, and dried or candied fruits is sparse. Freire and Offord [57] isolated *Cronobacter* spp. (formerly *Enterobacter sakazakii*) from cashew and Brazilian nuts. The obtained results showing the lack of *Cronobacter* spp. in seeds are inconsistent with data published by Hochel et al. [58], who detected them in 41.2% of the 34 analyzed samples of seeds, and isolated *C. sakazakii*, *C. turicensis*, and *C. malonaticus*.

## 3. Materials and Methods

### 3.1. Sample Collection

The research material consisted of 64 samples of market food products, i.e.:

samples of different types of nut: Italian and Brazil nuts, hazelnuts, almonds, pecan, cashew, pine nuts, and macadamia (20 samples);samples of dried fruit: prunes, raisins, cherries, sour cherries, figs, banana, dates, apricot, black currant, cranberry, Goji and Chia berries (24 samples);samples of candied fruit: mango, pineapple, Jackfruit, plums, and passion fruit (8 samples);samples of seed: sunflower and pumpkin seeds (4 samples);samples of mixes of seed, dried fruit and nut (mixes in various proportions of: raisins, dried cranberries, walnuts, hazelnuts, cashews, almonds, and sunflower seeds) (8 samples).

The samples were bought in stores in Warsaw (Poland) at different supermarkets, from September 2018 to February 2019. They originated from seven producers. All samples were packed in unitary packages weighing from 50 g to 250 g. No “sold by weight” products were analyzed in this study. The samples were transported at ambient temperature and immediately tested in the laboratory. All samples were examined within their “best before” date and were analyzed in three replicates.

### 3.2. Microbiological Analysis of Samples

All samples were determined for:

total plate count of bacteria (TPC) on the PCA medium (Merck, Warsaw, Poland); incubation conditions 72 h at 30 °C [59];yeasts and molds counts on the YGC medium (Merck, Warsaw, Poland); incubation conditions 5 days at 25 °C [60];occurrence of bacteria from the genus *Cronobacter* [53].

For the analysis of TCB, yeasts and molds counts, a 10 g test portion was taken from each sample and diluted with 90 mL of Ringer’s solution. After soaking for 15 min, the mixture was homogenized (Stomacher 80, Seward, Worthing, UK). Serial dilutions were also prepared in Ringer’s solution. In this study, all analyses were done in duplicate.

### 3.3. Phenotypic Identification of Presumptive Cronobacter spp. Isolates

The detection of *Cronobacter* was as according to the ISO standard 22964:2017 [53]. Typical colonies were selected from the chromogenic agar, purified on TSA (Oxoid, Poznań, Poland), and biochemically characterized. Biochemical features (according to the ISO 22964: 2017 standard [53]) were determined in all isolates. *Cronobacter* isolates included those that were oxidase negative, are able to hydrolyse 4-nitrophenyl α-D-glucopyranoside substrate, to produce acid from sucrose, also those that are unable to dexarboxylase L-lysine, and to produce acid from d-sorbitol.

### 3.4. Genetic Identification of Cronobacter spp. Isolates

#### 3.4.1. Extraction of DNA

Template DNA was extracted from isolates cultivated on TSA agar using the method described by Yamagishi et al. [61]. Bacterial colonies were suspended in 500 µL sterile DN-ase free water. The suspensions were boiled in a water bath (10 min), cooled on ice and centrifuged at 12,000× *g* for 10 min. Supernatant was transferred into a new tube and used as a template DNA. The amount of DNA was determined using the Thermo Scientific NanoDrop^TM^ 1000 Spectrophotometer (Thermo-Fisher Scientific Inc., Warsaw, Poland).

#### 3.4.2. PCR-RFLP

The 659-bp fragment of the *rpo*B gene was amplified using CroF and CroR primers, according to Li et al. [24]. The PCR reaction were performed as described by Berthold-Pluta et al. [26]. The 659-bp amplicon was purified (GenElute PCR Clean-Up Kit, Sigma-Aldrich, Warsaw, Poland) and digested separately with restriction endonuclease *Hin*P1I and with a combination of *Hin*P1I and *Csp*6I restrictases (Fermentas, Thermo-Fisher Scientific Inc., Warsaw, Poland). Digestion reaction was performed in a reaction mixture containing 350 ng of the purified amplicon; 2 μL 10× Thermo Scientific Tango Buffer; 10 U of *Hin*P1I enzyme (single digestion) or 8 U of *Hin*P1I and 16 U of *Csp*6I enzyme (double digestion), and sterile nuclease-free water up to 20 μL. After 2 h of incubation at temperature 37 °C, the enzyme was thermally inactivated at a temperature of 65 °C for 20 min. The digested fragments were separated by agarose gel electrophoresis (4%, m/v, at 90 V, TBE buffer). The PCR-RFLP profiles were visualized under UV light (VersaDoc System and QuantityOne 4.4.0 software; Bio-Rad, Warsaw, Poland). GeneRuler^TM^ 50 bp DNA Ladder (Fermentas, Thermo-Fisher Scientific Inc., Warsaw, Poland) was applied as a molecular marker to determine the size of DNA restriction fragments. Species identification was based on a comparison of the restriction patterns with the theoretical profiles predicted in *in-silico* analysis with the *rpo*B gene sequence of reference strains obtained from GenBank (Table 7).

#### 3.4.3. RAPD-PCR

Intra-species differentiations were carried out using three different RAPD-PCRs with primers: RP, UBC245 and UBC282, as described in our previous study [26]. 1000 bp/1 kb DNA ladder (GeneOn) was used as a molecular size standard. The RAPD patterns were analyzed using the VersaDoc System (Bio-Rad) and QuantityOne 4.4.0 software (Bio-Rad). All reactions were carried out in two independent replications.

### 3.5. Statistical Analysis of Results

The obtained results were subjected to a statistical analysis using Statistica version 13 software (TIBCO Software Inc., Kraków, Poland, 2017). One-way analysis of variance (ANOVA) was conducted. The Tukey test was applied to compare the significance of differences between mean values at a significance level of α = 0.05.

## 4. Conclusions

In this study, almost 8% of the analyzed samples, including samples of dried fruits, seeds and mixes of dried fruits, nuts and seeds, were disqualified for human consumption due to excessive counts of mold (>3 log CFU g^−1^), which fell short of meeting the guidelines for products of this type adopted and implemented in international fora. This level of contamination may pose risks to consumer health due to the potential presence of mycotoxins. *Cronobacter* species were not detected in any of the analyzed samples of dried fruits, candied fruits, and seeds. The prevalence of *Cronobacter* spp. in retail nuts and in mixes of dried fruits, seeds and nuts accounted for 50.0% and 25.0%, respectively. Three *Cronobacter* species: *C. malonaticus*, *C. turicensis* and *C. sakazakii,* were isolated from the samples of nuts (almonds, hazelnuts, cashew, pine nuts, macadamia, and Brazilian), and mixes of nuts, dried fruits and seeds. The presence of these three species, which are the only pathogens in the *Cronobacter* genus, in RTE foods can pose a potential risk to human health, especially for the seniors or persons with some previous pathology. Conclusively, risk assessment due to the consumption of RTE plant products by these groups of consumers must be analyzed by respective epidemiological surveillance agencies.

## Figures and Tables

**Figure 1 pathogens-10-00900-f001:**
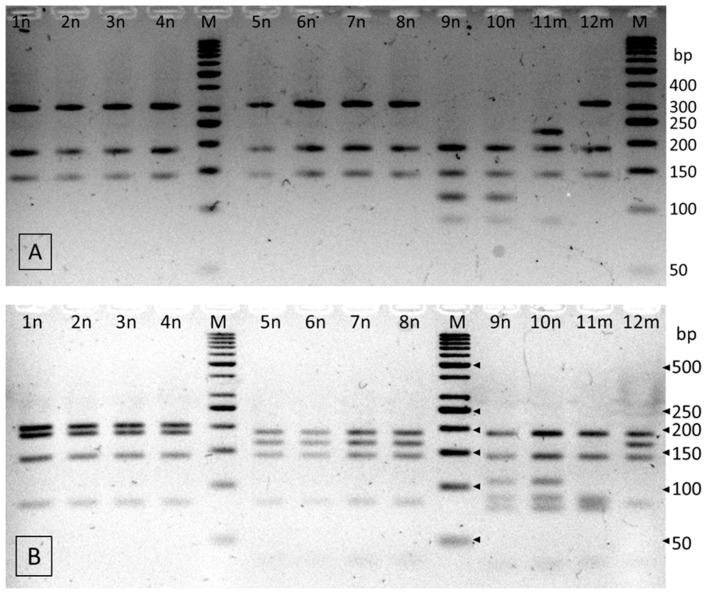
RFLP patterns of the PCR-amplified *rpoB* gene fragment of 12 *Cronobacter* spp. isolates digested with a single restrictase *Hin*P1I (**A**) and with a combination of *Hin*P1I and *Csp*6I restriction enzymes (**B**). M—molecular size standard, GeneRuler™ 50 bp DNA Ladder.

**Figure 2 pathogens-10-00900-f002:**
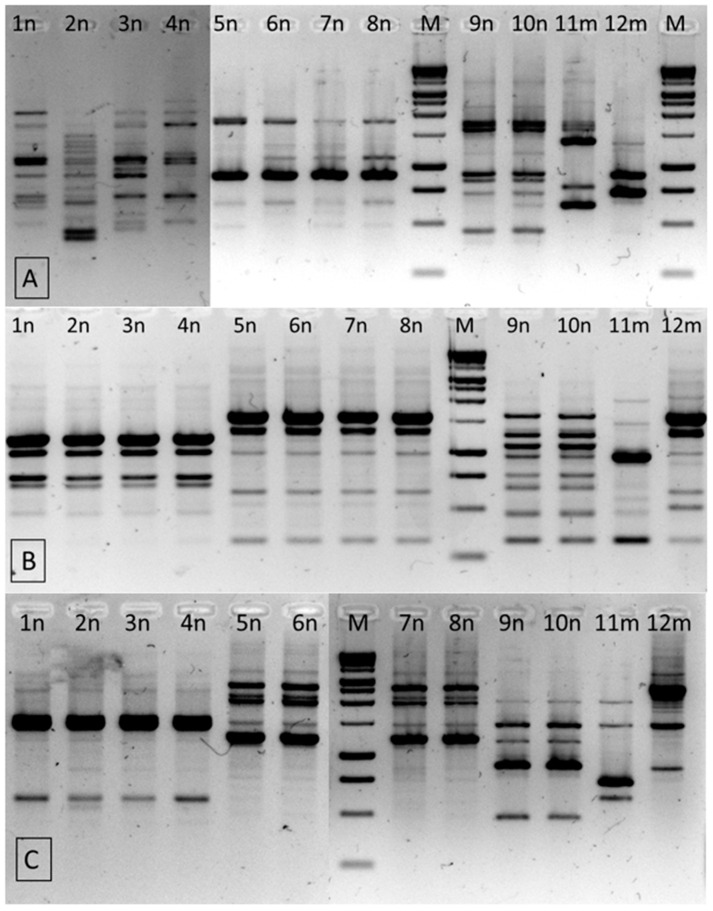
RAPD-PCR (Random Amplified Polymorphic DNA by PCR) patterns of 12 *Cronobacter* spp. isolates with primer RP (**A**), UBC245 (**B**) and UBC 282 (**C**). M- molecular size standard, 1000 bp/1 kb DNA ladder.

**Table 1 pathogens-10-00900-t001:** Total plate count of bacteria in the analyzed samples.

Food Products (Number of Analyzed Samples)	Number of Samples							
	Total Plate Count of Bacteria [log Colony Forming Units (CFU) g^−1^]							
	Not Detected in 0.1 g	> 1–2	> 2–3	> 3–4	> 4–5	> 5–6	Range	Mean ± Standard Deviation
Nuts (20)	4	5	4	5	2	0	1.2–4.4	2.80 ab ± 1.03
Dried fruits (24)	10	2	6	4	0	2	1.9–5.3	3.07 b ± 1.06
Candied fruits (8)	0	3	5	0	0	0	1.5–2.8	2.20 a ± 0.46
Seeds (4)	0	0	2	1	1	0	2.4–4.4	3.17 b ± 0.80
Mixes of dried fruits, seeds and nuts (8)	1	2	3	2	0	0	1.4–3.2	2.47 ab ± 0.62
Total (64)	15	12	20	12	3	2	1.2–5.3	2.26 ± 1.37

a, b—means with different letters in a column are significantly different (*p* < 0.05, *n* = 6).

**Table 2 pathogens-10-00900-t002:** Total count of yeasts in the analyzed samples.

Food Products (Number of Analyzed Samples)	Number of Samples				
	Total Count of Yeasts [log CFU g^−1^]				
	Not Detected in 0.1 g	>1–2	>2–3	Range	Mean ± Standard Deviation
Nuts (20)	16	3	1	1.4–2.6	1.88 a ± 0.47
Dried fruits (24)	23	1	0	1.8	1.80 a ± 0.00
Candied fruits (8)	8	0	0	–	–
Seeds (4)	4	0	0	–	–
Mixes of dried fruits, seeds and nuts (8)	6	2	0	1.5–1.9	1.70 a ± 0.22
Total (64)	57	6	1	1.4–2.6	1.82 ± 0.37

a—means with different letters in a column are significantly different (*p* < 0.05, *n* = 6).

**Table 3 pathogens-10-00900-t003:** Total count of molds in the analyzed samples.

Food Products (Number of Analyzed Samples)	Number of Samples					
	Total Count of Molds [log CFU g^−1^]					
	Not Detected in 0.1 g	>1–2	>2–3	>3–4	Range	Mean ± Standard Deviation
Nuts (20)	2	8	10	0	1.3–2.6	1.99 a ± 0.41
Dried fruits (24)	8	4	10	2	1.8–4.0	2.49 b ± 0.59
Candied fruits (8)	7	1	0	0	1.5	1.50 a ± 0.01
Seeds (4)	0	2	1	1	1.0–3.8	2.13 ab ± 1.09
Mixes of dried fruits, seeds and nuts (8)	4	0	2	2	2.7–3.8	3.22 c ± 0.48
Total (64)	21	15	23	5	1.0–4.0	2.29 ± 0.69

a–c—means with different letters in a column are significantly different (*p* < 0.05, *n* = 6).

**Table 4 pathogens-10-00900-t004:** PCR-RFLP (Polymerase Chain Reaction - Restriction Fragments Length Polymorphism) analysis of the *rpoB* gene (657/9 bp) of 12 *Cronobacter* spp. isolates.

RFLP-Pattern	Strain	Fragment Length (bp) *	Species Identification	References
A	1n, 2n, 3n, 4n	312, 186, 142 (*HinP*1I) 206, 186, 142, 83 (*HinP*1I + *Csp*6I)	*C. turicensis*	*C. turicensis* LMG 23827
B	5n, 6n, 7n, 8n, 12m	312, 186, 142 (*HinP*1I) 186, 167, 142, 83 (*HinP*1I + *Csp*6I)	*C. malonaticus*	*C. malonaticus* LMG 23826
C	9n, 10n	186, 142, 114, 110, 88 (*HinP*1I) 186, 142, 106, 88, 79 (*HinP*1I+ *Csp*6I)	*C. sakazakii*	*C. sakazakii* NTCT 8155
D	11m	224, 186, 142, 88 (*HinP*1I) 186, 142, 88, 83, 79 (*HinP*1I+ *Csp*6I)	*C. sakazakii*	*C. sakazakii* ATCC 29544

* All fragments smaller than 50 bp were not visible on the gel and thus, were excluded from comparison.

**Table 5 pathogens-10-00900-t005:** Genetic differentiation of 12 *Cronobacter* spp. isolates from tested products.

Isolate No.	Origin	Species Identification by PCR-RFLP	PCR-RFLP Pattern	RAPD-PCR * Patterns		
				RP	UBC245	UBC282
1n	almonds	*C. turicensis*	A	1	1	1
2n	hazelnuts	*C. turicensis*	A	2	1	1
3n	almonds	*C. turicensis*	A	1	1	1
4n	almonds	*C. turicensis*	A	1	1	1
5n	hazelnuts	*C. malonaticus*	B	3	2	2
6n	cashew nuts	*C. malonaticus*	B	3	2	2
7n	pini nuts	*C. malonaticus*	B	3	2	2
8n	macadamia nuts	*C. malonaticus*	B	3	2	2
9n	Brazilian nuts	*C. sakazakii*	C	4	3	3
10n	Brazilian nuts	*C. sakazakii*	C	4	3	3
11m	mixes of dried fruits, seeds and nuts	*C. sakazakii*	D	5	4	4
12m	mixes of dried fruits, seeds and nuts	*C. malonaticus*	B	6	5	5

* RAPD-PCR–Random Amplified Polymorphic DNA by PCR.

**Table 6 pathogens-10-00900-t006:** Presence of bacteria from the genus *Cronobacter* in the analyzed samples.

Food Product	Presence of Bacteria from the Genus *Cronobacter ** [Positive Samples Count/Analyzed Samples Count (% of Positive Samples)]	Presence of *C. sakazakii *** [Positive Samples Count/Analyzed Samples Count (% of Positive Samples)]	Presence of *C. turicensis *** [Positive Samples Count/Analyzed Samples Count (% of Positive Samples)]	Presence of *C. malonaticus *** [Positive Samples Count/Analyzed Samples Count (% of Positive Samples)]
Nuts	10 ^1^/20 (50.0)	2 ^2^ /20 (10.0)	4 ^3^/20 (20.0)	4 ^4^ /20 (20.0)
Dried fruits	0/24 (0.0)	0/24 (0.0)	0/24 (0.0)	0/24 (0.0)
Candied fruits	0/8 (0.0)	0/8 (0.0)	0/8 (0.0)	0/8 (0.0)
Seeds	0/4 (0.0)	0/4 (0.0)	0/4 (0.0)	0/4 (0.0)
Mixes of dried fruits, seeds and nuts	2 ^1^/8 (25.0)	1 ^2^/8 (12.5)	0/8 (0.0)	1 ^4^/8 (12.5)
Total	12/64 (18.8)	3/64 (4.7)	4/64 (6.3)	5/64 (7.8)

* based on ISO-22964:2017 [53]; ** based on the results of PCR-RFLP analysis of the *rpoB* gene with *Hin*P1l and *Csp*6l restriction enzymes; ^1^ presence of *Cronobacter* spp. in samples of: almonds, hazelnuts, Brazilian, cashew, pini and macadamia nuts, and mixes of dried fruits, seeds and nuts; ^2^ presence of *C. sakazakii* in two samples of Brazilian nuts and one sample of mixed dried fruits, seeds and nuts; ^3^ presence of *C. turicensis* in three samples of almonds and one sample of hazelnuts; ^4^ presence of *C. malonaticus* in four samples of: hazelnuts, cashew, pini, macadamia nuts, and one sample of mixed dried fruits, seeds and nuts.

**Table 7 pathogens-10-00900-t007:** The predicted sizes of fragment length cleaved by *HinP*1I and *Csp*6I restriction endonuclease.

Species	GenBank Accession No.	Predicted Fragments in In Silico Analysis (bp)
*C. sakazakii* ATCC 29544	FJ717657	186, 142, 106, 88, 79, 39, 15, 2
*C. sakazakii* NTCT 8155	CP012253	186, 142, 88, 83, 79, 35, 23, 15, 4, 2
*C. malonaticus* LMG 23826	CP013940	186, 167, 142, 83, 39, 23, 15, 2
*C. turicensis* LMG 23827	NC_013282	206, 186, 142, 83, 23, 15, 2
*C. dubliniensis* LMG 23823	CP012266	211, 167, 142, 87, 35, 15, 2
*C. muytjensii* ATCC 51329	CP012268	188, 167, 96, 87, 46, 35, 23, 15, 2
*C. condimenti* LMG 26250	CP012264	211, 167, 142, 122, 15, 2
*C. universalis* NCTC 9529	CP012257	167, 157, 142, 83, 39, 31, 23, 15, 2

## Data Availability

The data presented in this study are available on request from the corresponding author.

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
