# Peer review of "Microbiological Quality of Nuts, Dried and Candied Fruits, Including the Prevalence of Cronobacter spp."

_pathogens, 2021, doi:10.3390/pathogens10070900_

Round 1

Reviewer 1 Report

General comments:

In the original manuscript, the author was advised to take care of some comments.

I reviewed all the comments in my previous report, and I found that this version included all the corrections.

In general, this manuscript has a valuable topic. The manuscript is well written except for minor English language check required. The experimental design is adequate. The current version of this manuscript has improved from the original just a couple of minor corrections.

Simple corrections needed:

1- Please change Results obtained were subjected to a statistical analysis. to

    The obtained results were subjected to a statistical analysis.

2- Table 2 please add a full stop at the end of the table title and at the end of the table footnote.

     The same comment for table 4 and table 5.

3-    Line 340 Please change Almost 8 % of the samples analyzed in this study, including these of dried fruits to .. In this study, almost 8 % of the analyzed samples, including samples of dried fruits.

4- Line 354 should be analyzed to must be analyzed

Conclusions:

The author was advised to add this section to the original manuscript, and I see that it is present in the current version. This section was organized and well written. The author provided a good conclusion for the study and includes the significant findings. and it was supported by the results. Finally, the author included in the last part of the conclusion a valuable recommendation based on the results from this study.

References:

The authors provided enough citations, and it was UpToDate.

**Considering all the hard work and the valuable data in this manuscript, I am convinced that this manuscript is very valuable and will be suitable to be published in the pathogen journal after a minor revision.

Author Response

Dear Reviewer 1,

Once again, we would like to thank you for your contribution to improving our manuscript.

We hope that revised manuscript will be evaluated as improved, in any case, we are willing to consider any further request.

All changes compared to the previous version have been highlighted in green.

Simple corrections needed:

1- Please change Results obtained were subjected to a statistical analysis. to

    The obtained results were subjected to a statistical analysis.

Corrected.

2- Table 2 please add a full stop at the end of the table title and at the end of the table footnote.

     The same comment for table 4 and table 5.

Corrected.

3-    Line 340 Please change Almost 8 % of the samples analyzed in this study, including these of dried fruits to .. In this study, almost 8 % of the analyzed samples, including samples of dried fruits.

Corrected.

4- Line 354 should be analyzed to must be analyzed

Corrected.

Reviewer 2 Report

Abstract:

line 19: change "total count of bacteria (TCB)" to "total plate count "TPC" and throughout manuscript

Line 23: change "The occurrence of cronobacters" to "The presence/absence of Cronobacter species..." - do not use "cronobacters" but rather "Cronobacter species"

Introduction:

Line 37: "change "causes the belief" to "prevents growth of many pathogens and lends to the microbiological safety of the LMFs"

Line 42: change to "Nuts are used by the food industry as they are associated with a healthy, convenient, and ..."

Line 44: Chante "The data published" to "Data published..."

Line 51: natural, non-pathogenic, and pathogenic are overlapping...unless the authors suggest that non-pathogenic microorganisms are different than the natural flora found on these products?

Lines 62-65: change as only 7 species recognized (sakazakii, malonaticus, turicensis, universalis, condimenti, dublinensis, and muytjensii)

Materials and Methods:

Question: What is the actual method used for isolating Cronobacter species? it appears the ISO method was used to confirm but did authors use the ISO method? Same for section 2.3 - where did isolates come from (i.e., how were they isolated)?

Section 2.4: RAPD is not commonly used. Why did authors not consider differentiation via cgcA approach (Carter et al, 2013. Applied and Environmental Microbiology 79: 734)? This appears to be a better speciation.  Was WGS not available to do sub-typing?

Results and Discussion:

Line 261: yellow pigmentation is no longer used for profiling Cronobacter species

Table 3. Two C. sakazakii reference strains can question how many reference strains need to be used to accurately assess the PCR-RFLP approach? In fact, the different bands might also be useful to replacing the RAPD approach?

Line 284: how many times was the RAPD protocol done on each isolate? The Figure 2 has many faint bands that prevent accurate subtyping. How many biological replicates were done on each strain? Technical replicates?

Author Response

Dear Reviewer 2

Thank you very much for reviewing our manuscript: Microbiological quality of nuts, dried and candied fruits, including the prevalence of Cronobacter spp. We have adopted all your suggestions.

Your suggestions have seriously contributed to the improvement of our manuscript. All changes compared to the original version have been highlighted in blue. Hope the revised manuscript will be evaluated as improved, in any case, we are willing to consider any further request.

Abstract:

line 19: change "total count of bacteria (TCB)" to "total plate count "TPC" and throughout manuscript

Corrected.

Line 23: change "The occurrence of cronobacters" to "The presence/absence of Cronobacter species..." - do not use "cronobacters" but rather "Cronobacter species"

Corrected.

Introduction:

Line 37: "change "causes the belief" to "prevents growth of many pathogens and lends to the microbiological safety of the LMFs"

Corrected.

Line 42: change to "Nuts are used by the food industry as they are associated with a healthy, convenient, and ..."

Corrected.

Line 44: Chante "The data published" to "Data published..."

Corrected

Line 51: natural, non-pathogenic, and pathogenic are overlapping...unless the authors suggest that non-pathogenic microorganisms are different than the natural flora found on these products?

Corrected.

Lines 62-65: change as only 7 species recognized (sakazakii, malonaticus, turicensis, universalis, condimenti, dublinensis, and muytjensii)

Corrected.

Materials and Methods:

Question: What is the actual method used for isolating Cronobacter species? it appears the ISO method was used to confirm but did authors use the ISO method? Same for section 2.3 - where did isolates come from (i.e., how were they isolated)?

The detection of Cronobacter was as according to the ISO standard 22964:2017 [51]. Typical colonies were selected from the chromogenic agar, purified on a tryptone soya agar plates - TSA (Oxoid, Poland), and biochemically characterized

Section 2.4: RAPD is not commonly used. Why did authors not consider differentiation via cgcA approach (Carter et al, 2013. Applied and Environmental Microbiology 79: 734)? This appears to be a better speciation.  Was WGS not available to do sub-typing?

We do agree that the cgcA approach are useful tool for Cronobacter species identification, however this multiplex PCR not include Cronobacter condimenti species. We isolated this species in our previous study (Berthold-Pluta et al., 2017). The in silico RFLP analysis of rpoB gene sequence of C. condimenti strain 1330 (JQ316670) and our unpublished data indicate that PCR-RFLP of the rpoB gene of C. condimenti give restriction pattern specific only for C. condimenti species. Therefore, we chose this method as a good tool for species differentiation of these bacteria. RAPD analysis were performed for intraspecies differentiation of strains, as WGS was unfortunately not available due to cost and technical limitation. However, for better discrimination we used three different primers.

Results and Discussion:

Line 261: yellow pigmentation is no longer used for profiling Cronobacter species

Yes we agree that yellow pigmentation is no longer used to profiling Cronobacter species, however we wanted to mention that our isolates showed it.

Table 3. Two C. sakazakii reference strains can question how many reference strains need to be used to accurately assess the PCR-RFLP approach? In fact, the different bands might also be useful to replacing the RAPD approach?

Our study as well as other literature data indicate that PCR-RFLP assay is species specific. However, we observed the mutation in rpoB gene of some C. sakazakii isolates that leads to the formation of an additional cleavage site for HinP1I enzyme. This generates a new restriction pattern characteristic for some strains within the C. sakazaki species. Other unspecified RFLP pattern has not been identified.

In the present study, the results of PCR-RFLP and RAPD-PCR are not fully consistent. Five C. malonaticus isolates had the same RFLP-PCR patterns, but a different RAPD-PCR pattern was obtained for one of the  isolates, 12m. The results of our previous work also indicate the higher discrimination power of RAPD-PCR (Berthold-Pluta et al., 2017). Therefore, we used RFLP-PCR only for species identification and preliminary differentiation of isolates. We found that RAPD-PCR approach is more suitable for typing Cronobacter spp. isolates.

Line 284: how many times was the RAPD protocol done on each isolate? The Figure 2 has many faint bands that prevent accurate subtyping. How many biological replicates were done on each strain? Technical replicates?

All RAPD-PCR assays were performed in two independent replicates (line 187) and the results of both analysis were consistent. All isolates, on the basis of the obtained genotypic patterns, were grouped in the same in both replicates. We excluded fr
